# Learning High-Order Relationships of Brain Regions

## Abstract

Discovering reliable and informative interactions among brain regions from functional magnetic resonance imaging (fMRI) signals is essential in neuroscientific predictions of cognition. Most of the current methods fail to accurately characterize those interactions because they only focus on pairwise connections and overlook the high-order relationships of brain regions. We delve into this problem and argue that these high-order relationships should be *maximally informative and minimally redundant* (MIMR). However, identifying such high-order relationships is challenging and highly under-explored. Methods that can be tailored to our context are also non-existent. In response to this gap, we propose a novel method named HYBRID that aims to extract MIMR high-order relationships from fMRI data. HYBRID employs a CONSTRUCTOR to identify hyperedge structures, and a WEIGHTER to compute a weight for each hyperedge. HYBRID achieves the MIMR objective through an innovative information bottleneck framework named *multi-head drop-bottleneck* with theoretical guarantees. Our comprehensive experiments demonstrate the effectiveness of our model. Our model outperforms the state-of-the-art predictive model by an average of $12.1\%$, regarding the quality of hyperedges measured by CPM, a standard protocol for studying brain connections.

## 1 Introduction

Discovering reliable and informative relations among brain regions using fMRI signals is crucial for predicting human cognition in and understanding brain functions (Kucian et al., 2008; 2006; Li et al., 2015b;a; Satterthwaite et al., 2015; Wang et al., 2016).

However, despite the clear multiplexity of the brain's involvement in cognition (Logue & Gould, 2014; Barrasso-Catanzaro & Eslinger, 2016; Knauff & Wolf, 2010; Reineberg et al., 2022), the current imaging biomarker detection methods (Shen et al., 2017; Gao et al., 2019; Li et al., 2021) focus only on the contributing roles of the pairwise connectivity edges. In contrast, most brain functions involve distributed patterns of interactions among multiple regions (Semedo et al., 2019). For instance, executive planning requires the appropriate communication of signals across many distinct cortical areas (Logue & Gould, 2014). Pairwise connectivity, without considering the brain's high-order structure, can lead to inconsistent findings across studies with low prediction performance. Although there are few works (Zu et al., 2016; Xiao et al., 2019; Li et al., 2022) working on discovering the high-order relationships of brain regions, they cannot effectively extract meaningful patterns in terms of cognition since 1) the discovering process of high-order relationships is not guided by cognitive targets; 2) the weak expressiveness of traditional machine learning methods (e.g. lasso regression) hinders the accurate understanding of brain region interactions (Cao et al., 2022; Richards et al., 2019).

In this paper, we aim to explore the identification of high-order relationships through deep learning fashion. Our objective is to learn high-order relationships that are *maximally informative and minimally redundant* (MIMR): maximizing the predictive performance of the high-order relationships toward a neurological outcome while diminishing the participation of unrelated brain regions. Such a criterion endows the model with the capacity to identify more succinct and interpretable structures (Yu et al., 2020; Miao et al., 2022a;b), which benefits the understanding of human brains. A formal definition of the MIMR criterion could be found in Equation 8 from an information bottleneck view.

We formulate high-order relationships as hyperedges in a hypergraph. In this context, regions are treated as nodes. Unlike a traditional graph where edges connect only two nodes, a hypergraph allows edges, known as hyperedges, to connect any number of nodes. The hypergraph should be weighted, and the weights of hyperedges are considered as strengths of high-order relationships, which contain the information relevant to cognition (Figure 1).

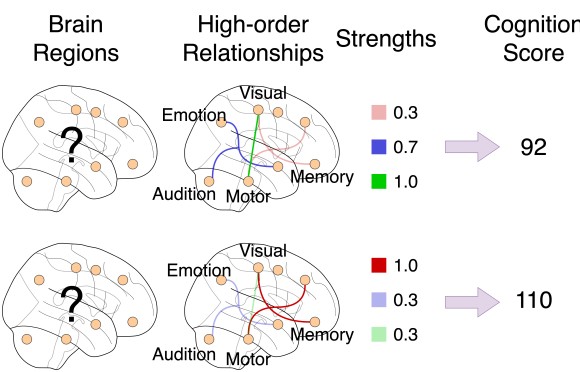

Figure 1: We identify high-order relationships of brain regions, where hyperedges possess strong relevance to cognition (maximal informativeness). Meanwhile, they contain the least irrelevant information.

However, current methods for hypergraph construction, which mostly are based on neighbor distances and neighbor reconstruction (Wang et al., 2015; Liu et al., 2017; Jin et al., 2019; Huang et al., 2009), are unsuitable in our context for several reasons: 1) they are unable to learn MIMR hyperedges due to the absence of a tractable objective for learning such hyperedges. 2) they fall short of learning consistent structures across subjects, which contradicts the belief that the cognitive function mechanism of healthy human beings should be similar (Wang et al., 2023). 3) the number of hyperedges is restricted to the number of nodes, which may lead to sub-optimal performance. Furthermore, although information bottleneck (IB) is a prevailing solution to learn MIMR representations in deep learning (Kim et al., 2021; Alemi et al., 2016; Luo et al., 2019), existing IB methods focus on extracting compressed representations of inputs instead of identifying underlying structures such as hypergraphs. Harnessing the IB framework for identifying hypergraphs necessitates both architectural innovations and theoretical derivations.

**Proposed work** In this paper, we propose **Hy**pergraph of **B**rain **R**egions via mult**i**-head **D**rop-bottleneck (HYBRID), a novel approach for identifying maximally informative yet minimally redundant high-order relationships of brain regions. The overall pipeline of HYBRID is depicted in Figure 2. HYBRID is equipped with a CONSTRUCTOR and a WEIGHTER. The CONSTRUCTOR identifies the hyperedge structures of brain regions by learning sets of masks, and the WEIGHTER computes a weight for each hyperedge. To advance the IB principle for hyperedge identification, we further propose *multi-head drop-bottleneck* and derive its optimization objective.

HYBRID gets rid of searching in an exponential space through learning masks to identify hyperedges, which guarantees computational efficiency. Its feature-agnostic masking mechanism design ensures HYBRID to learn consistent structures across subjects. Moreover, the model is equipped with a number of parallel heads, and each head is dedicated to a hyperedge. Through this, HYBRID is able to identify any number of hyperedges, depending on how many heads it is equipped with. Additionally, the proposed *multi-head drop-bottleneck* theoretically guarantees the maximal informativeness and minimal redundancy of the identified hyperedges.

We evaluate our methods on 8 datasets of human subjects under different conditions and at different participation times. We quantitatively evaluate our approach by a commonly used protocol for studying brain connections, CPM (Shen et al., 2017) (Appendix B), and show that our model outperforms the state-of-the-art deep learning models by an average of 12.1% on a comprehensive benchmark. Our post-hoc analysis demonstrates that hyperedges of higher degrees are considered more significant, which indicates the significance of high-order relationships in human brains.

## 2 PROBLEM DEFINITION & NOTATIONS

**Input** We consider ABCD, which is a widely used benchmark in fMRI studies (5.1). This dataset is a collection of instances, where each instance corresponds to an individual human subject. An instance is represented by the pair $(X, Y)$. $X \in \mathbb{R}^{N \times d}$ represents the features for each subject, where $N$ is the number of brain regions and $d$ is the number of feature dimensions. The features are obtained by connectivity, consistent with previous works (5.1). $Y \in \mathbb{R}$ denotes the prediction

target, such as fluid intelligence. Section 5 elaborates more details about datasets and preprocessing procedures. For how to obtain features of regions from raw data, see Paragraph 5.1 for more details.

**Goal** Based on the input $X$, HYBRID aims to learn a weighted hypergraph of the brain, where regions are nodes. To achieve this, HYBRID identifies a collection of hyperedges $H = (\boldsymbol{h}^1, \boldsymbol{h}^2, \cdots, \boldsymbol{h}^K)$, and assigns weights $\boldsymbol{w} = [w^1, w^2, \cdots, w^K]^T$ for all hyperedges. These hyperedges and their weights, which represent strengths of hyperedges, are expected to be the most informative (i.e. relevant to the target $Y$) yet the least redundant.

**Representation of hyperedges** As mentioned before, we use $H$ to denote the collection of hyperedge structures and $\boldsymbol{h}^k$ to denote the $k$-th hyperedge. We use the following representation for a hyperedge:

$$\boldsymbol{h}^k = \boldsymbol{m}^k \odot X \in \mathbb{R}^{N \times d} \tag{1}$$

where $\boldsymbol{m}^k \in \{0, 1\}^N$ is a mask vector and $\odot$ denotes broadcasting element-wise multiplication. In other words, each $\boldsymbol{h}^k$ is a random row-zeroed version of $X$.

## 3 RELATED WORK

**Hypergraph construction** Existing hypergraph construction methods are mostly based on neighbor reconstruction and neighbor distances. For example, the $k$ nearest neighbor-based method (Huang et al., 2009) connects a centroid node and its $k$ nearest neighbors in the feature space to form a hyperedge. Wang et al. (2015); Liu et al. (2017); Jin et al. (2019); Xiao et al. (2019) further refine these neighbor connections through various regularization. However, the number of hyperedges of these methods is restricted to the number of nodes, and hyperedges obtained by these methods are inconsistent across instances. Zhang et al. (2022; 2018) proposed to iteratively refine a noisy hypergraph, which is obtained by the aforementioned methods. Therefore, they share the same limitations as the aforementioned methods. In addition, these methods are unable to learn MIMR hyperedges due to the absence of a tractable objective. Other methods such as attributed-based methods (Huang et al., 2015; Joslyn et al., 2019) are ill-adapted to our context since they require discrete labels or a prior graph topology. Different from these methods, we provide a way to learn a consistent hypergraph through a deep-learning model without any prior topology. Furthermore, thanks to the proposed *multi-head drop-bottleneck*, these hyperedges are theoretically ensured MIMR.

**High-order relationships in fMRI** Although some methods are working on high-order relationships in fMRI, they are limited or inconsistent with our MIMR objective. Xiao et al. (2019); Li et al. (2022) used the existing non-learning-based hypergraph construction methods, which may lead to noisy and inexpressive hypergraphs. Zu et al. (2016); Santoro et al. (2023) enumerated all hyperedges with degrees lower than 3, which can only discover a tiny portion of all possible hyperedges in exponential space and is not scalable to a large degree. Rosas et al. (2019) proposed O-information, which reflects the balance between redundancy and synergy. The O-information metric is utilized by Varley et al. (2023) to study fMRI data. However, the objective of these methods is not consistent with ours: although both of us are quantifying the redundancy of high-order relations, our method is to learn those that are most informative toward a cognition score, while theirs is to depict the synergy and redundancy within a system.

**Information bottleneck** Information bottleneck (IB) (Tishby et al., 2000) is originally a technique in data compression. The key idea is to extract a summary of data, which contains the most relevant information to the objective. Alemi et al. (2016) first employed an IB view of deep learning. After that, IB has been widely used in deep learning. The applications span areas such as computer vision (Luo et al., 2019; Peng et al., 2018), reinforcement learning (Goyal et al., 2019; Igl et al., 2019), natural language processing (Wang et al., 2020) and graph learning (Yu et al., 2020; 2022; Xu et al., 2021; Wu et al., 2020). Unlike these studies that use IB to extract a compressed representation or a select set of features, our approach focuses on identifying the underlying structures of the data.

**Connectivity-based cognition prediction** Recently, deep learning techniques have been increasingly employed in predicting cognition based on the connectivity of brain regions. A substantial portion of these studies (Ahmedt-Aristizabal et al., 2021; Li et al., 2019; Cui et al., 2022b; Kan

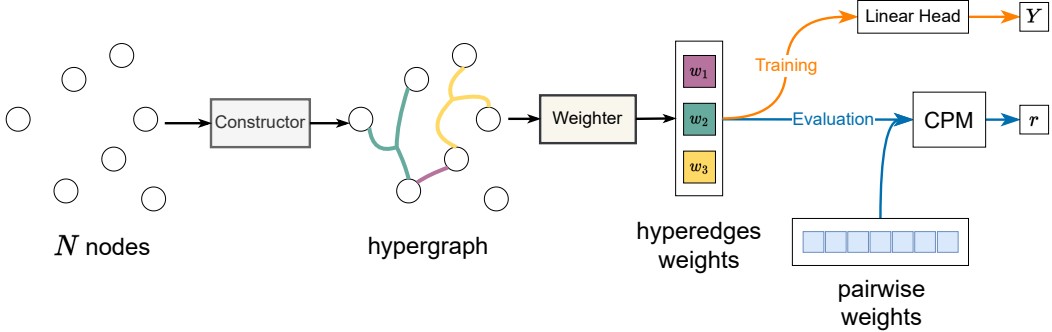

Figure 2: Overview of the HYBRID pipeline when the total number of hyperedges $K = 3$. hyperedge are in distinct colors for clarity. The CONSTRUCTOR identifies hyperedges in the hypergraph, where regions are nodes. The WEIGHTER computes a weight for each hyperedge. These weights, representing strengths of hyperedges, are expected to be informative in terms of our target $Y$. There are two separate phases after obtaining weights of hyperedges: 1) Training. The model's parameters are trained under the supervision of $Y$; 2) Evaluation. The output weights, as well as pairwise weights, are fed into the CPM (see Appendix B).

et al., 2022a; Cui et al., 2022a; Said et al., 2023) model the brain network as a graph, in which regions act as nodes and pairwise correlations form the edges. These methods predominantly utilize Graph Neural Networks (GNNs) to capture the connectivity information for predictions. In addition to GNNs, Kan et al. (2022b) proposed to use transformers with a specially designed readout module, leveraging multi-head attention mechanisms to capture pairwise connectivity. However, these methods heavily rely on pairwise connectivity and neglect more intricate higher-order relationships. This oversight, on the one hand, leads to sub-optimal prediction performances; and on the other hand, prevents domain experts from acquiring insightful neuroscience interpretations, given that human cognition often involves multiple regions.

## 4 METHOD

**Method Overview** HYBRID consists of a CONSTRUCTOR $\mathcal{F}_c$, a WEIGHTER $\mathcal{F}_w$, and a LINEARHEAD $\mathcal{F}_l$. At a high level, the CONSTRUCTOR $\mathcal{F}_c$ is responsible for identifying hyperedges $H$ from the data to construct the hypergraph. After that, the WEIGHTER $\mathcal{F}_w$ calculates a weight for each hyperedge. Finally, based on all the weights $\boldsymbol{w}$, the LINEARHEAD $\mathcal{F}_l$ predicts the label $Y$. An illustration of this pipeline is shown in Figure 2. The pipeline can be formulated as

$$X \xrightarrow[\mathcal{F}_c]{} H \xrightarrow[\mathcal{F}_w]{} \boldsymbol{w} \xrightarrow[\mathcal{F}_l]{} Y \tag{2}$$

We will elaborate on all of the details of the architecture below.

### 4.1 LEARNING THE HYPERGRAPH BY MULTI-HEAD MASKING

Given an instance, represented by $X = [X_1, X_2, \cdots, X_N]^T \in \mathbb{R}^{N \times d}$, where $X_i \in \mathbb{R}$ is a column vector representing the features of region $i$. These regions are nodes in the hypergraph we are going to construct. Hyperedges in the hypergraph can be beneficial in the learning task because it is essential to model the interactions between more than two regions.

**Hyperedges construction** In this paragraph, we elaborate how the CONSTRUCTOR identifies the hyperedges, i.e. $H = \mathcal{F}_c(X)$.

Suppose the number of hyperedges is $K$, which is a predefined hyperparameter. We assign a head to each hyperedge. Each head is responsible for constructing a hyperedge by selecting nodes belonging to that hyperedge.

Specifically, to construct the $k$-th hyperedge, the CONSTRUCTOR's $k$-th head outputs a column vector $\boldsymbol{m}^k \in \{0,1\}^N$, where each element in the vector corresponds to a region.

$$\boldsymbol{m}^k = [\mathbb{1}(p_{\theta,1}^k), \mathbb{1}(p_{\theta,2}^k), \cdots, \mathbb{1}(p_{\theta,N}^k)]^T \in \{0,1\}^N \tag{3}$$

where $p_{\theta,i}^k, i = 1, 2, \cdots, N$ are learnable parameters. $\mathbb{1} : [0,1] \mapsto \{0,1\}$ is an indicator function, which is defined as $\mathbb{1}(x) = 1$ if $x > 0.5$ and $\mathbb{1}(x) = 0$ if $x \leq 0.5$. And $\boldsymbol{m}^k$ is a column vector corresponding to the $k$-th hyperedge. Note that since there is no gradient defined for the indicator operation, we employ the stopgradient technique (Oord et al., 2017; Bengio et al., 2013) to approximate the gradient.

In the vector $\boldsymbol{m}^k$, 0 indicates nodes being masked out, and 1 indicates nodes not being masked. Nodes that are not masked are considered to form a hyperedge together. We use $\boldsymbol{h}^k$ to represent the masked version of $X$

$$\begin{aligned} \boldsymbol{h}^k &= \boldsymbol{m}^k \odot X \\ &= [\boldsymbol{m}_1^k X_1, \boldsymbol{m}_2^k X_2, \cdots, \boldsymbol{m}_N^k X_N] \in \mathbb{R}^{N \times d} \end{aligned} \tag{4}$$

where $\odot$ is the broadcast element-wise multiplication. $\boldsymbol{m}_j^k$ is the $j$-th element of the vector $\boldsymbol{m}^k$.

We obtain $K$ hyperedges for $K$ sets of masks. We use $H$ to denote the collection of all hyperedges.

$$H = (\boldsymbol{h}^1, \boldsymbol{h}^2, \cdots, \boldsymbol{h}^K) \tag{5}$$

**Hyperedge weighting** After obtaining the structure (i.e. member nodes) of each hyperedge, the WEIGHTER will calculate each hyperedge's weight, which is supposed to indicate the importance of that hyperedge, based on the member nodes and their features, i.e. $\boldsymbol{w} = \mathcal{F}_w(H)$.

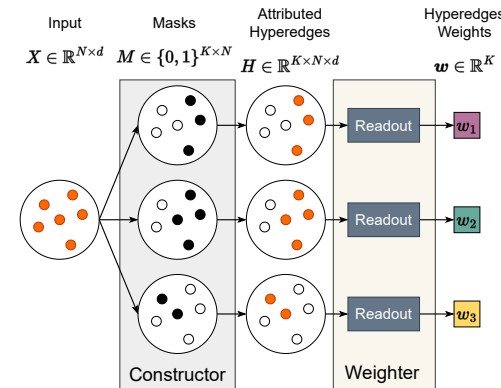

Figure 3: Architecture details of the CONSTRUCTOR and the WEIGHTER when the number of nodes $N = 6$ and the number of hyperedges $K = 3$. The CONSTRUCTOR learns the hyperedge structure by masking nodes. The WEIGHTER computes the weight of each hyperedge based on the remaining nodes and their features.

These weights are obtained by a $\mathrm{Readout}$ module, which is composed of: 1) summing over all the non-masked nodes feature-wisely; 2) dim reduction operation.

$$w^k = \mathrm{Readout}(\boldsymbol{h}^k) = \mathrm{DimReduction}(\boldsymbol{m}^{k^T}\boldsymbol{h}^k) \in \mathbb{R} \tag{6}$$

where $w^k$ is the weight of the $k$-th hyperedge and $\mathrm{DimReduction}$ is an MLP with ReLU activations, where the output dimension is 1. For all hyperedges, we obtain $K$ hyperedges in total, $\boldsymbol{w} = [w^1, w^2, \cdots, w^K]^T \in \mathbb{R}^K$. Finally, these weights will be fed into the final linear head to predict the label of the instance,

$$\hat{Y} = \mathcal{F}_l(\boldsymbol{w}) \in \mathbb{R} \tag{7}$$

In contrast to previous hypergraph construction methods (Jin et al., 2019; Xiao et al., 2019), which identify hyperedges by refining neighbors and assign weights by aggregating node features, HYBRID makes these procedures learnable and thus is able to identify MIMR hyperedges in a data-driven way through expressive neural networks. We decide the number of hyperedges $K$ according to the study in Appendix E.2.

## 4.2 OPTIMIZATION FRAMEWORK

We propose a new IB framework named *multi-head drop-bottleneck* to optimize HYBRID. To adopt an information bottleneck view of HYBRID, we consider $X$, $Y$ and $H$ are random variables in the Markovian chain $X \leftrightarrow Y \leftrightarrow H$. According to our MIMR objective, we optimize

$$\arg \max I(H; Y) - \beta I(H; X) \tag{8}$$

where $\beta$ is a coefficient trading off informativeness and redundancy. Since optimizing the mutual information for high-dimensional continuous variables is intractable, we instead optimize the lower

bound of Equation 8. Specifically, for the first term (informativeness), it is easy to show

$$
\begin{aligned}
I(H;Y) &= \mathbb{H}[Y] - \mathbb{H}[Y|H] \\
&= \mathbb{H}[Y] + \mathbb{E}_{p(Y,H)}[\log p(Y|H)] \\
&= \mathbb{H}[Y] + \mathbb{E}_{p(Y,H)}[\log q_\phi(Y|H)] + \mathbb{E}_{p(H)}[\mathrm{KL}(p(Y|H)|q_\phi(Y|H))] \\
&\geq \mathbb{H}[Y] + \mathbb{E}_{p(Y,H)}[\log q_\phi(Y|H)]
\end{aligned}
\tag{9}
$$

where $\mathbb{H}[\cdot]$ is the entropy computation. Since there is no learnable component in the entropy of $Y$, we only need to optimize the second term $\mathbb{E}_{p(Y,H)}[\log q_\phi(Y|H)]$. $q_\phi$ can be considered as a model that predicts $Y$ based on $H$, which essentially corresponds to $\mathcal{F}_l \circ \mathcal{F}_w$, where $\circ$ is the function composition. In practice, we set $q_\phi$ as a Gaussian model with variance 1 as most probabilistic machine learning models for continuous data modeling. (Alemi et al., 2016; Luo et al., 2019; Peng et al., 2018; Kingma & Welling, 2013).

For the second term (redundancy) in Equation 8, we have

**Proposition 1.** *(Upper bound of $I(H;X)$ in multi-head drop-bottleneck)*

$$
I(H;X) \leq \sum_{k=1}^{K} I(\boldsymbol{h}^k; X) \leq \sum_{k=1}^{K} \sum_{i=1}^{N} I(\boldsymbol{h}_i^k; X_i) = \sum_{k=1}^{K} \sum_{i=1}^{N} \mathbb{H}[X_i](1 - p_{\theta,i}^k)
\tag{10}
$$

where $\boldsymbol{h}_j^i$ and $X_j$ is the $j$-th row of $\boldsymbol{h}^i$ and $X$ respectively. $p_{\theta,i}^k$ is the mask probability in Equation 3. $\mathbb{H}$ is the entropy computation. The equality holds if and only if nodes are independent and hyperedges do not overlap. The second inequality is inspired by Kim et al. (2021). The proof of the proposition can be found in Appendix A.

Therefore, instead of optimizing the intractable objective 8, we optimize its upper bound (i.e. loss function) according to Equation 9 and 10.

$$
\mathcal{L} = \|Y - \mathcal{F}_l \circ \mathcal{F}_w \circ \mathcal{F}_c(X)\|_2^2 + \beta \sum_{k=1}^{K} \sum_{i=1}^{N} \mathbb{H}[X_i](1 - p_{\theta,i}^k) \geq -I(H,Y) + \beta I(H,X)
\tag{11}
$$

The learnable components are the shallow embeddings in Equation 3, the DimReduction MLP in Equation 6 and the LinearHead $\mathcal{F}_l$ in Equation 7. For how do we choose the trade-off coefficient $\beta$, see Appendix E.3 for more discussions.

## 4.3 Computational Complexity

Suppose we have $N$ regions as $N$ nodes and $K$ hyperedges. For the Constructor, the unignorable computation is from the mask operation. The computational complexity of this step for each hyperedge is $O(Nd)$ since it does a pairwise multiplication operation of a matrix of size $N \times d$. Given that there are $K$ hyperedges, the total complexity is $O(NKd)$. For the Weighter, the computation is from the dim reduction operation and the linear head. The dim reduction operation is an MLP. In this work, the hidden dimensions are a fraction of the original feature dimension. Therefore, the complexity of the dim-reduction MLP is $O(d^2)$. The linear head only contributes $O(d)$, which is neglectable. As a result, the computational complexity of the whole model is $O(NKd + d^2) = O(N^2K)$ since the feature dimension is equal to the number of regions (5.1). This complexity is just at the same scale as that of MLPs even though we are addressing a more challenging task: identifying high-order relationships in an exponential space.

## 5 Experiments

In this section, we conducted experiments to validate the quality of the learned hyperedges regarding the cognition phenotype outcome. Furthermore, we conducted ablation studies to validate the key components in our model. We also analyzed our results both quantitatively and qualitatively.

## 5.1 Experiment Settings

**Metric**  To evaluate the quality of hyperedges obtained by Hybrid, we use CPM (Shen et al., 2017), a standard model that could evaluate the correlation between connectivity and the prediction

target, due to its high impact in the community. In the original implementation of CPM, weights of pairwise edges are obtained by Pearson correlation between nodes. These weights, as pairwise connectivity, are fed into the CPM. CPM will output a metric that measures the overall correlation between edges and the prediction target, which can be considered as a measure of edge qualities. This process is formulated as

$$r' = \text{CPM}(\boldsymbol{w}_p, Y) \tag{12}$$

where $\boldsymbol{w}_p \in \mathbb{R}^{K_p}$ denotes the pairwise edge weights and $K_p$ is the total number of pairwise edges. $r'$ is a metric that measures the quality of weights based on positive and negative correlations to the cognition score. To evaluate the quality of the learned weights for our model and baselines, we concatenate the learned weights $\boldsymbol{w}_h \in \mathbb{R}^{K_h}$ with the pairwise ones together into the CPM, and thus adjust Equation 12 to

$$r = \text{CPM}([\boldsymbol{w}_p \| \boldsymbol{w}_h], Y) \tag{13}$$

where $[\cdot \| \cdot]$ denotes the concatenation of two vectors. The $r$ value reflects the quality of learned weights in terms of the prediction performance since it measures the overall correlation between weights and the prediction target. In our model, $\boldsymbol{w}_h = \boldsymbol{w}$, which is the learned hyperedge weights.

**Dataset**  We use functional brain imaging data from the first release of the Adolescent Brain Cognitive Development (ABCD) study, collected from $11,875$ children aged between 9 to 10 years old (Casey et al., 2018). The functional MRI (fMRI) data is collected from children when they were resting and when they performed three emotional and cognitive tasks. The fMRI data is processed using BioImage Suite (Joshi et al., 2011) with the standard preprocessing procedures, such as slice time and motion correction, registration to the MNI template (see details in Greene et al. (2018) and Horien et al. (2019)). We delete scans with more than 0.10 mm mean frame-to-frame displacement. We use the ABCD imaging data collected from the baseline (release 2.0) as well as the 2-year follow-up (release 3.0). For each time point, we included children subjected to four conditions: the resting state where the brain is not engaged in any activity (Rest), the emotional n-back task (EN-back), the Stop Signal task (SST), and the Monetary Incentive Delay (MID) task. In conclusion, we obtain 8 datasets from 2 timepoints and 4 conditions (Rest, SST, EN-back, MID). Statistics of each dataset are summarized in Appendix C.

For the prediction target, we consider fluid intelligence as our label. Fluid intelligence reflects the general mental ability and plays a fundamental role in various cognitive functions. Recent literature has seen success in predicting intelligence based on pairwise predictive modeling (Dubois et al., 2018a;b), and we aim to improve the prediction accuracy of the current methods.

**Data preprocessing**  The raw fMRI data of an instance is represented in four dimensions (3 spatial dimensions + 1 temporal dimension), which can be imagined as a temporal sequence of 3D images. First, brain images are parceled into regions (or nodes) using the AAL3v1 atlas (Rolls et al., 2020). Following previous works (Kan et al., 2022b; Li et al., 2021; Thomas et al., 2022), each region's time series is obtained by averaging all voxels in that region. Consistent with previous connectivity-based methods (Li et al., 2021; Kan et al., 2022b; Ktena et al., 2018; Said et al., 2023), for each region, we use its Pearson correlation coefficients to all regions as its features. We randomly split the data into train, validation, and test sets in a stratified fashion. The split ratio is 8:1:1.

**Baselines**  We compare our method with 3 classes of baselines: 1) *standard* method, which is exactly the classical method that predicts outcomes based on pairwise edges (Shen et al., 2017; Dadi et al., 2019; Wang et al., 2021). The comparison with standard methods shows whether the high-order connectivity has its advantage over the classical pairwise one or not. 2) *hypergraph construction* methods. We consider $k$NN (Huang et al., 2009), $l_1$ hypergraph (Wang et al., 2015), and $l_2$ hypergraph (Jin et al., 2019). 3) Connectivity-based *cognition prediction* methods, which are state-of-the-art predictive models based on brain connectivity. We consider BrainNetGNN (Mahmood et al., 2021), BrainGNN (Li et al., 2021), and BrainNetTF (Kan et al., 2022b). BrainGB (Cui et al., 2022a) is a study of different brain graph neural network designs and we include its best design as a baseline. Note that since these models are unable to identify hyperedge structures of brain regions, we input their last layer embeddings (each entry as a weight) into the CPM model. Note that our weights $\boldsymbol{w}$ are also last layer embeddings in HYBRID.

| Type | Model | SST 1 | EN-back 1 | MID 1 | Rest 1 | SST 2 | EN-back 2 | MID 2 | Rest 2 |
|---|---|---|---|---|---|---|---|---|---|
| Standard | pairwise | 0.113 | 0.218 | 0.099 | 0.164 | 0.201 | 0.322 | 0.299 | 0.289 |
| Hypergraph Construction | kNN | 0.115 | 0.268 | 0.168 | 0.127 | 0.257 | 0.266 | 0.238 | 0.315 |
| | $l_1$ hypergraph | 0.099 | 0.223 | 0.125 | 0.126 | 0.145 | 0.295 | 0.242 | 0.259 |
| | $l_2$ hypergraph | $0.096_{\pm 0.002}$ | $0.197_{\pm 0.003}$ | $0.118_{\pm 0.003}$ | $0.157_{\pm 0.016}$ | $0.203_{\pm 0.005}$ | $0.272_{\pm 0.004}$ | $0.289_{\pm 0.011}$ | $0.307_{\pm 0.006}$ |
| Connectivity based Prediction | BrainNetGNN | $0.227_{\pm 0.060}$ | $0.287_{\pm 0.043}$ | $0.266_{\pm 0.046}$ | $0.221_{\pm 0.040}$ | $0.468_{\pm 0.058}$ | $0.480_{\pm 0.068}$ | $0.506_{\pm 0.057}$ | $0.453_{\pm 0.028}$ |
| | BrainGB | $0.190_{\pm 0.073}$ | $0.214_{\pm 0.051}$ | $0.265_{\pm 0.048}$ | $0.176_{\pm 0.066}$ | $0.447_{\pm 0.089}$ | $0.483_{\pm 0.077}$ | $0.458_{\pm 0.064}$ | $0.432_{\pm 0.076}$ |
| | BrainGNN | $0.262_{\pm 0.030}$ | $0.235_{\pm 0.032}$ | $0.260_{\pm 0.049}$ | $0.185_{\pm 0.058}$ | $0.455_{\pm 0.028}$ | $0.391_{\pm 0.077}$ | $0.445_{\pm 0.078}$ | $0.368_{\pm 0.041}$ |
| | BrainNetTF | $0.327_{\pm 0.084}$ | $0.338_{\pm 0.056}$ | $0.370_{\pm 0.098}$ | $\mathbf{0.334_{\pm 0.084}}$ | $0.633_{\pm 0.178}$ | $0.631_{\pm 0.142}$ | $0.629_{\pm 0.123}$ | $0.588_{\pm 0.138}$ |
| Ours | HYBRID | $\mathbf{0.361_{\pm 0.058}}$ | $\mathbf{0.348_{\pm 0.061}}$ | $\mathbf{0.386_{\pm 0.060}}$ | $0.223_{\pm 0.056}$ | $\mathbf{0.738_{\pm 0.054}}$ | $\mathbf{0.714_{\pm 0.037}}$ | $\mathbf{0.816_{\pm 0.053}}$ | $\mathbf{0.730_{\pm 0.049}}$ |

Table 1: $r$ values of our hyperedges compared to baselines. Results are averaged over 10 runs. Deterministic methods do not have standard deviations.

| Model | SST 1 | EN-back 1 | MID 1 | Rest 1 | SST 2 | EN-back 2 | MID 2 | Rest 2 |
|---|---|---|---|---|---|---|---|---|
| HYBRID | $\mathbf{0.361_{\pm 0.058}}$ | $\mathbf{0.348_{\pm 0.061}}$ | $\mathbf{0.386_{\pm 0.060}}$ | $0.223_{\pm 0.056}$ | $\mathbf{0.738_{\pm 0.054}}$ | $\mathbf{0.714_{\pm 0.037}}$ | $\mathbf{0.816_{\pm 0.053}}$ | $\mathbf{0.730_{\pm 0.049}}$ |
| HYBRID$_{\text{NoMask}}$ | $0.297_{\pm 0.035}$ | $0.274_{\pm 0.057}$ | $0.323_{\pm 0.059}$ | $0.221_{\pm 0.034}$ | $0.653_{\pm 0.036}$ | $0.599_{\pm 0.059}$ | $0.757_{\pm 0.021}$ | $0.543_{\pm 0.038}$ |
| HYBRID$_{\text{RndMask}}$ | $0.256_{\pm 0.069}$ | $0.191_{\pm 0.046}$ | $0.255_{\pm 0.080}$ | $0.190_{\pm 0.051}$ | $0.541_{\pm 0.069}$ | $0.514_{\pm 0.038}$ | $0.598_{\pm 0.064}$ | $0.482_{\pm 0.083}$ |
| HYBRID$_{\text{SoftMask}}$ | $0.343_{\pm 0.042}$ | $0.314_{\pm 0.040}$ | $0.320_{\pm 0.055}$ | $\mathbf{0.245_{\pm 0.061}}$ | $0.707_{\pm 0.042}$ | $0.662_{\pm 0.058}$ | $0.796_{\pm 0.031}$ | $0.655_{\pm 0.030}$ |

Table 2: Ablation studies on the masking mechanism. Results are averaged over 10 runs.

**Implementation & Training Details**  Due to the data scarcity, training on individual datasets would result in serious overfitting. To mitigate this, we train our model as well as baselines using all eight datasets together. Details of the overfitting issue and our parameter-sharing strategy are further discussed in Appendix E.1. See Appendix D for other implementation and training details such as hyperparameter choices and software/hardware specifications.

## 5.2 QUALITY OF HYPEREDGES

We report $r$ values by CPM in Table 1. As we can see, HYBRID outperforms the state-of-the-art predictive models on 7 datasets, with an average improvement of $12.1\%$. Moreover, HYBRID outperforms all other deep learning baselines in efficiency, with $87\%$ faster than the next fastest one (BrainNetTF). Refer to Appendix F for more runtime details.

**Ablation Studies**  We conduct an ablation study on the effect of our masking mechanism. Specifically, we compare our model with 3 variants: 1) HYBRID$_{\text{RndMask}}$: Replace the learnable masks with randomized masks with the same sparsity at the beginning of training. 2) HYBRID$_{\text{NoMask}}$: Do not mask at all, which means all nodes and their features are visible to each head. 3) HYBRID$_{\text{SoftMask}}$: Remove the indicator function and use $p_{\theta,i}^k$ directly in Equation 3. Ablation results are shown in Table 2. We find the original HYBRID and the HYBRID$_{\text{SoftMask}}$ outperform all other variants, which demonstrates the effect of learnable masks. Moreover, the original HYBRID is better than its soft version HYBRID$_{\text{SoftMask}}$, which demonstrates our sparse and succinct representations reserve more relevant information than smooth ones. Other ablation studies such as the choices of the number of hyperedges and choices of $\beta$ can be found in Appendix E.

## 5.3 FURTHER ANALYSIS

**Hyperedge degree distribution**  We plot the hyperedge degree distribution in Fig.4a. We find there are two distinct clusters in the figure. The first cluster is hyperedges with degree $\leq 5$. 1-degree and 2-degree hyperedges are special cases of our method: 1-degree hyperedges are individual nodes, which imply the contribution of individual regions to the cognition. 2-degree hyperedges reveal the importance of traditional pairwise connectivity. The other cluster concentrates around degree 25, which implies the importance of relationships of multiple regions.

**Hyperedges with higher degree are more significant**  CPM conducts a significance test on pairwise edges and hyperedges internally based on a linear regression model, and thus we can obtain a P-value for each hyperedge from the significance test. We define the significance of a hyperedge as $1 - P_v \in [0, 1]$ where $P_v$ is the P-value of that hyperedge.

The relationship between hyperedge degree and its significance is shown in Fig 4b. In this figure, we find a strong positive correlation between a hyperedge's degree and its significance, which indicates

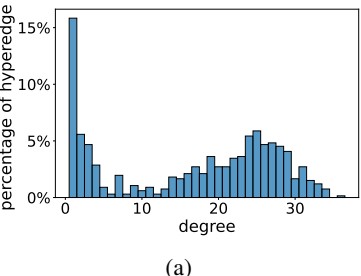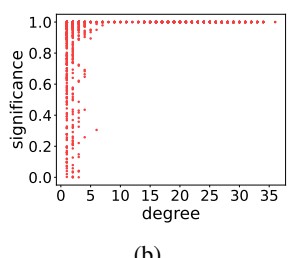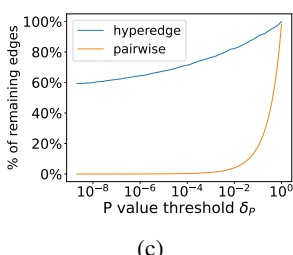

(a)                     (b)                     (c)

Figure 4: Hyperedge profiles. **(a)** Hyperedge degree distribution of learned hyperedges. **(b)** Correlation between hyperedge degree and significance. **(c)** Comparison between the number hyperedges and pairwise edges under different significance thresholds. The total number of hyperedges is 32. And the total number of pairwise edges is $26,896$.

that interactions of multiple brain regions play more important roles in cognition than pairwise or individual ones. It is also worth mentioning that there is a turning point around degree 5, which corresponds to the valley around 5 in Figure 4a.

**Comparison with pairwise edges**  To compare the significance in cognition between pairwise edges and learned hyperedges, we plot the number of remaining edges under different thresholds in Figure 4c. We find out that the learned hyperedges are much more significant than pairwise ones. Also note that even if we set the threshold to an extremely strict value ($1 \times 10^{-8}$), there are still $60\%$ hyperedges considered significant. This evidence shows that our high-order relationships are much more significant than the traditional pairwise connectivity, which implies relationships involving multiple brain regions could be much more essential in cognition.

**Region importance**  To better understand the roles of each brain region in cognition under different conditions, we studied the frequency at which each region appears in a hyperedge out of all identified hyperedges. The frequency, which can be considered as a measure of region importance, is visualized in Figure 5. Visual regions (*Fusiform*, *Cuneus*, *Calcarine*) are especially active due to the intensive visual demands in all three conditions. We found that the *Frontal_Inf_Oper* and *ACC_pre*, recognized for their participation in response inhibition (Porn-

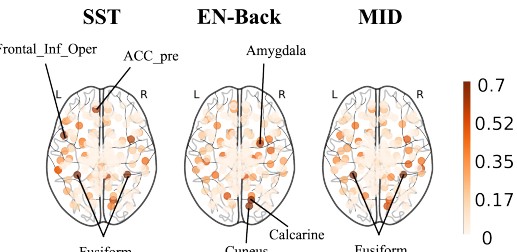

Figure 5: Visualization of the frequency of each region under different conditions.

pattananangkul et al., 2016), frequently appear in the SST task. This aligns with what the SST task was designed to test. Interestingly, of the three conditions (SST, EN-back, MID), only EN-back prominently involves the *Amygdala*, a key region for emotion processing. This makes sense as EN-back is the only condition related to emotion. More visualizations and interpretations on the resting-state region importance and individual hyperedges can be found in Appendix G.

## 6 CONCLUSION

In this work, we propose HYBRID for identifying maximally informative yet minimally redundant (MIMR) high-order relationships of brain regions. To effectively optimize our model, we further proposed a novel information bottleneck framework with theoretical guarantees. Our method outperforms state-of-the-art models of hypergraph construction and connectivity-based prediction. The result analysis shows the effectiveness of our model. We expect such advancements could benefit clinical studies, providing insights into neurological disorders, and offering improved diagnostic tools in neurology.

**Limitations**  HYBRID only considers static high-order relations. Given that ABCD tasks are dynamic, including temporal changes and interactions, it will be interesting to study the evolution of these high-order relationships.

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

## A   PROOF OF THE UPPER BOUND

In this section, we prove the upper bound of $I(H; X)$ in *multi-head drop-bottleneck* in Equation 10.

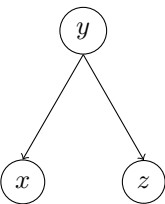

Figure 6: The graphical model of random variables $X$, $Y$ and $Z$.

**Lemma 1.** *Given random variables $X$, $Y$ and $Z$. Their relationships are described in the graphical model illustrated in Figure 6. We have*

$$I(X; Y|Z) \leq I(X; Y) \tag{14}$$

*Proof.*

$$
\begin{aligned}
I(X; Y|Z) - I(X; Y) &= \int p(x, y, z) \log \frac{p(z)p(x, y, z)}{p(x, z)p(y, z)} \mathrm{d}x\mathrm{d}z\mathrm{d}y \\
&\quad - \int p(x, y) \log \frac{p(x, y)}{p(x)p(y)} \mathrm{d}x\mathrm{d}y \\
&= \int p(x, y, z) \log \frac{p(z)p(x, y, z)}{p(x, z)p(y, z)} \mathrm{d}x\mathrm{d}z\mathrm{d}y \\
&\quad - \int p(x, y, z) \log \frac{p(x, y)}{p(x)p(y)} \mathrm{d}x\mathrm{d}y\mathrm{d}z \\
&= \int p(x, y, z) \log \frac{p(z)p(x, y, z)p(x)p(y)}{p(x, z)p(y, z)p(x, y)} \mathrm{d}x\mathrm{d}z\mathrm{d}y \\
&= \int p(x, y, z) \log \frac{p(z)p(x, z|y)p(y)p(x)p(y)}{p(x, z)p(y, z)p(x, y)} \mathrm{d}x\mathrm{d}z\mathrm{d}y \\
&= \int p(x, y, z) \log \frac{p(z)p(x|y)p(z|y)p(y)p(x)p(y)}{p(x, z)p(y, z)p(x, y)} \mathrm{d}x\mathrm{d}z\mathrm{d}y \\
&= \int p(x, y, z) \log \frac{p(x)p(z)}{p(x, z)} \mathrm{d}x\mathrm{d}z\mathrm{d}y \\
&= \int p(x, z) \log \frac{p(x)p(z)}{p(x, z)} \mathrm{d}x\mathrm{d}z \\
&= -I(X; Z) \leq 0
\end{aligned}
\tag{15}
$$

which finishes the proof.

$\square$

**Corollary 1.1.** *Given the same graphical model 6, we have*

$$I(X, Z; Y) \leq I(X; Y) + I(Z; Y) \tag{16}$$

*Proof.* Using the chain rule of mutual information, we obtain

$$I(X, Z; Y) = I(X; Y|Z) + I(Z; Y) \tag{17}$$

According to Lemma 1, we have $I(X; Y|Z) \leq I(X; Y)$, which finishes the proof.

$\square$

**Theorem 2.** *For random variables in Equation 10, we have*

$$I(H; X) \leq \sum_{k=1}^{K} I(\boldsymbol{h}^k; X) \tag{18}$$

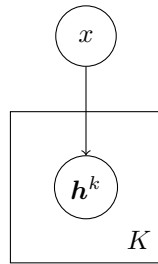

Figure 7: The graphical model of random variables $\boldsymbol{h}^k$ and $X$

*Proof.* According to the definitions of $X$ and $H$, which are described in Section 4, we can draw a graphical model of them in Figure 7. Define a new random variable $\boldsymbol{h}^{k_1:k_2} = [\boldsymbol{h}^{k_1}, \boldsymbol{h}^{k_1+1}, \cdots, \boldsymbol{h}^{k_2-1}, \boldsymbol{h}^{k_2}]$, which is a concatenation from $\boldsymbol{h}^{k_1}$ to $\boldsymbol{h}^{k_2}$. According to Corollary 1.1 we have

$$\begin{aligned} I(H; X) &\leq I(\boldsymbol{h}^1, X) + I(\boldsymbol{h}^{2:K}, X) \\ &\leq I(\boldsymbol{h}^1, X) + I(\boldsymbol{h}^2, X) + I(\boldsymbol{h}^{3:K}, X) \\ &\leq I(\boldsymbol{h}^1, X) + I(\boldsymbol{h}^2, X) + I(\boldsymbol{h}^3, X) + \cdots \\ &\leq \sum_{k=1}^{K} I(\boldsymbol{h}^k; X) \end{aligned} \tag{19}$$

$\square$

**Theorem 3.** *(proposition 1)*

$$I(H; X) \leq \sum_{k=1}^{K} \sum_{i=1}^{N} I(\boldsymbol{h}_i^k; X_i) = \sum_{k=1}^{K} \sum_{i=1}^{N} \mathbb{H}[X_i](1 - p_{\theta,i}^k) \tag{20}$$

*Proof.* Given **Theorem 2**. It suffices to prove

$$I(\boldsymbol{h}^k; X) \leq \sum_{i=1}^{N} I(\boldsymbol{h}_i^k; X_i) = \sum_{i=1}^{N} \mathbb{H}[X_i](1 - p_{\theta,i}^k), \quad \forall 1 \leq k \leq K \tag{21}$$

And this is exactly the conclusion in Kim et al. (2021) if we consider $\boldsymbol{h}^k$ and $X$ as $X$ and $Z$ respectively in their paper. $\square$

## B  CONNECTOME-BASED PREDICTIVE MODELING

Shen et al. (2017); Finn et al. (2015)) has shown tremendous promise in recent years in detecting imaging biomarkers. (Rosenberg et al., 2015; Dubois et al., 2018a; Rosenberg et al., 2020; 2016). Such a model, based chiefly on functional MRI data, can measure the significance of the input edge weights, which is revealed by a correlation coefficient that reflects the correlation between the edge weights and the neurological outcomes. One could expect a large correlation coefficient to indicate the high quality of edge weights. We utilize the CPM as an evaluation model to evaluate the quality of our learned hyperedges. Here is a pipeline overview of the CPM process:

1. **Connectivity Calculation**: For each subject, compute the Pearson correlation coefficients for each possible pair of brain regions. This is based on the fMRI data collected for those regions.

| Dataset | Rest 1 | SST 1 | EN-back 1 | MID 1 | Rest 2 | SST 2 | EN-back 2 | MID 2 |
|---|---|---|---|---|---|---|---|---|
| #(instances) | 1676 | 1673 | 1678 | 1678 | 1949 | 1053 | 1044 | 1062 |
| length of time series | 375 | 437 | 362 | 403 | 375 | 437 | 362 | 403 |

Table 3: Statistics of 8 datasets.

2. **Edge Significance**: Calculate the correlation between each brain connectivity edge and the outcome of interest (e.g., cognition scores) across all subjects. The correlation of an edge indicates its significance.
3. **Edge Selection**: Identify significant connectivity edges. These are the edges where the correlation values are greater than a predetermined significance threshold.
4. **Weight Summation**: For each subject, sum the weights of the significant edges identified in the previous step to derive a single summary score (scalar).
5. **Model Fitting**: Fit a linear model that predicts the neurological outcomes based on the summed weights, where each subject is a sample.
6. **Model Evaluation**: Across all subjects, calculate the correlation of predicted values and the neurological outcomes. Note that it is equivalent to the correlation between the summed weights and the outcomes, and is exactly the metric $r$ we use to evaluate our hyperedges in Equation 13.

Since positive edges and negative edges will cancel out with each other when being summed, we adopt the combining strategy in Boyle et al. (2023).

**CPM measures the quality of edge weights**    According to step 6, the evaluation of the predictive model could be measured by the correlation between predicted and ground-truth outcomes Shen et al. (2017). Since CPM is a linear model that predicts the outcome based on the sum of significant edge weights, the correlation is equal to the correlation between the sum and ground-truth outcomes (which is exactly the r in Eq. 13). Hence, one can expect a larger correlation if the edge weights are more correlated (and thus are more predictive).

**Significance of edges in CPM**    In step 2, CPM obtains a correlation coefficient $r^k$ for each edge weight $w^k$ and the cognition score $Y$ across all subjects. Consider a classical hypothesis test $H_0 : r^k = 0, H_1 : r^k \neq 0$. Assume $w^k$ and $Y$ are drawn from independent normal distribution (corresponds to $H_0$), the probability density function of correlation coefficient $r^k$ is

$$f(r^k) = \frac{\left(1 - r^{k^2}\right)^{n/2-2}}{\mathrm{B}\left(\frac{1}{2}, \frac{n}{2} - 1\right)},$$

where $n$ is the number of samples and $B$ is the beta function. Based on the distribution, we obtain a $P$-value for the $k$-th edge, which is used to measure the significance of the edge.

## C  DATASET DETAILS

The statistics of the number of instances and the time series length are summarized in Table 3.

## D  TRAINING DETAILS

**Hardware**    We train our model on a machine with a Intel Xeon Gold 6326 CPU and RTX A5000 GPUs.

**Software**    See Table 4 for software we used and the versions.

| notation | meaning | value |
|---|---|---|
| $lr$ | learning rate | $1 \times 10^{-3}$ |
| $K$ | number of hyperedges | 32 |
| $\beta$ | trade-off coefficients information bottleneck | 0.2 |
| $[h_1, h_2, h_3]$ | hidden sizes of the dim reduction MLP | $[32, 8, 1]$ |
| $B$ | batch size | 64 |

Table 5: Hyperparameter Choices

| Strategy ID | ShallowEmbeddings | DimReduction | LinearHead | Average Score |
|---|---|---|---|---|
| 1 | ✓ | ✓ | ✓ | 0.337 |
| 2 | ✓ | ✓ | | 0.335 |
| 3 | ✓ | | ✓ | 0.508 |
| 4 | ✓ | | | 0.539 |
| 5 | | ✓ | ✓ | 0.351 |
| 6 | | ✓ | | 0.384 |
| 7 | | | ✓ | 0.506 |
| 8 | | | | **0.558** |

Table 6: The table summaries the model's average performance under various parameter-sharing strategies. The symbol ✓ indicates a component being condition-specific, while an empty cell implies a component being shared across conditions.

| software | version |
|---|---|
| python | 3.8.13 |
| pytorch | 1.11.0 |
| cudatoolkit | 11.3 |
| numpy | 1.23.3 |
| ai2-tango | 1.2.0 |
| nibabel | 4.0.2 |

Table 4: Software versions

**Hyperparameter choices** The hyperparameters selection is shown in Table 5. Some crucial hyperparameters ablation experiments can be found in Appendix E.

# E  MORE ABLATION STUDIES

## E.1  PARAMETERS SHARING

As previously discussed, the scarcity of fMRI data can result in serious over-fitting. In our experiments, we observed that when the model is trained on each dataset individually, the validation set loss remains stagnant. To address this, we examined the model's performance under various parameter-sharing strategies. Specifically, we have three learnable components: the shallow embeddings in Equation 3, the dim reduction MLP in 6 and the LINEARHEAD $\mathcal{F}_l$. These components can be either shared or condition-specific. Results of different parameter-sharing strategy is shown in Table 6.

Note that strategy 1 is equivalent to training models separately for each condition. From the table, we find that the DimReduction component plays the most significant role. Making DimReduction condition-specific enhances the average performance by $10\%-20\%$. Strategy 8 is the best. However, domain experts may expect different structures under different conditions. Therefore, we still adopt condition-specific ShallowEmbeddings (i.e. strategy 4), given that their scores are really close. We argue that the fact that strategy 4 is not better than 8 is because our prediction target is fluid

intelligence, which is a general measurement of cognition and tends to be condition-agnostic. We anticipate a potential shift in results if we switch to a more condition-specific cognition phenotype outcome.

## E.2 Choices of number of hyperedges $K$

As explained in Section 4, we use $K$ heads for $K$ hyperedges. We study the correlation between the $r$ value and the number of hyperedges on three datasets:

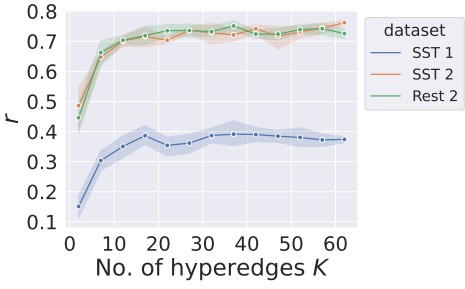 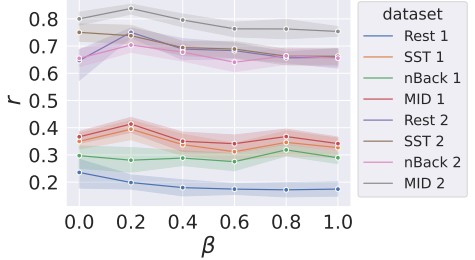

(a) The performance with increasing the number of hyperedges through 2, 7, 12, 17, 22, 27, 32, 37, 52, 57, 62 on three datasets.

(b) Performance with increasing $\beta$ from 0.1 to 1.0 with step 0.1 on all datasets.

From Figure 8a, we find that the overall performance increases dramatically before $K = 17$, but becomes stable and close to saturation after $K = 32$. To improve the efficiency while ensuring the performance, we choose $K = 32$.

## E.3 Choices of the trade-off coefficient $\beta$

In our optimization objective 8, $\beta$ acts as a trade-off parameter, which is a non-negative scalar that determines the weight given to the second term relative to the first. To study its fluence to the performance, we plot the model performances on all datasets under different $\beta$ in Figure 8b. We can see performances on 3 datasets (Rest 1, SST 2) consistently decrease when $\beta$ increases. However, on other 5 datasets (SST 1, MID 1, Rest 2, nBack 2, MID 2), we can observe a peak at $\beta = 0.2$. Accordingly, we adopt $\beta = 0.2$.

## F Runtime Comparison

Figure 9 summarizes the per-batch training time of all deep learning models. We find that HYBRID is the most efficient one, with 87% faster than the second one (BrainNetTF) and at least 1255% faster than the GNN-based ones.

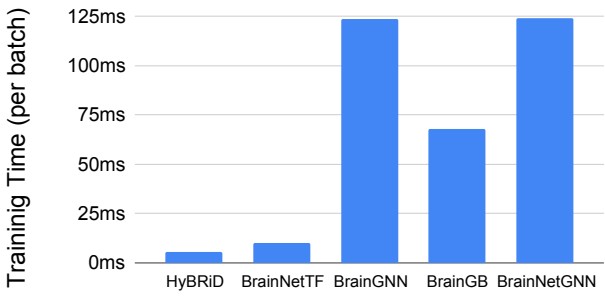

Figure 9: Training time per batch of all deep learning models.

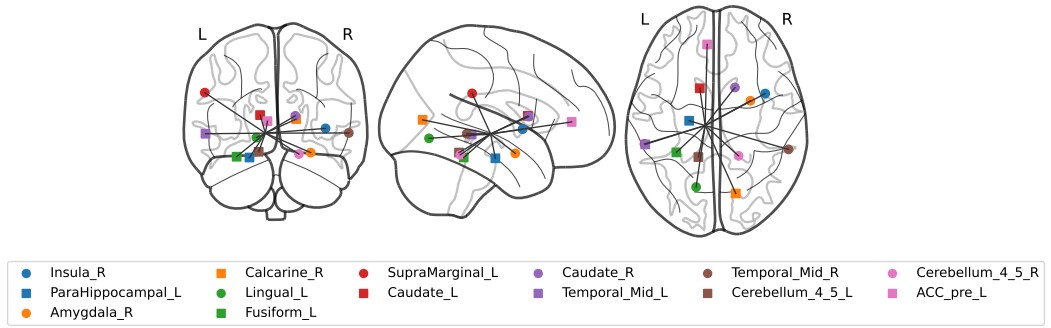

Figure 10: Visualization of the most significant hyperedge in EN-back condition

## G  MORE VISUALIZATIONS

**Hyperedge case study**   We visualize the most significant hyperedge under the EN-back condition. We observe a coordinated interaction of numerous brain regions, each fulfilling specific roles. Notably, some of these regions serve multi-functional purposes:

- **Memory Processing** *ParaHippocampal_L*, *Temporal Mid*: Essential for memory encoding and retrieval, these regions are integral to the EN-back task, facilitating the recall of previously viewed images.

- **Emotional processing** *Amygdala_R*: The amygdala is crucial for the processing of emotions, such as fear and pleasure. Since the EN-back task involves emotional stimuli, it is reasonable that the region is connected by the hyperedge.

- **Visual Processing**: *Calcarine_R*, *Lingual_L*, *Fusiform_L*. These regions are responsible for visual perception and some of them are related to complex visual contents like symbols and human faces, which were presented during the task.

- **Sensory** *SupraMarginal_L*: It is responsible for interpreting tactile sensors and perceiving limbs location. Its involvement is likely due to the requirement for participants to engage in specific physical actions, such as pressing buttons, during the task. *Temporal Mid*: It functions in multi-modal sensory integration.

- **Motor Control** *Cerebellum*: It is primarily responsible for muscle control. *Caudate*: It plays a crucial role in motor processes. Its involvement is likely attributed to participants engaging in physical actions, like pressing buttons.

- **Cognitive Control** *ACC_pre_L*: In the EN-back task, this region is likely crucial for maintaining focus, error detection and correction, conflict management in working memory, and modulating emotional responses to the task's demands.

**Resting-state brain region importance**   We visualize the region importance of the resting state in Figure 11. Different from task states, where specific brain regions are activated in response to particular tasks, brain activities during resting states, are not driven by external tasks, leading to more diffuse and less predictable patterns of activation. This makes it harder to pinpoint specific interactions or functions.

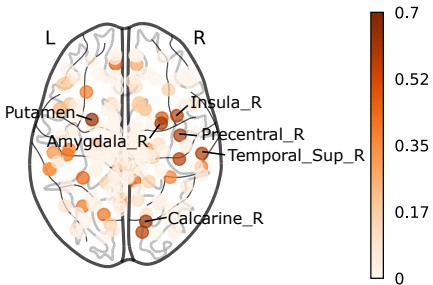

Figure 11: Region importance of resting state.

