# OpenReview forum: "Learning High-Order Relationships of Brain Regions"
_ICLR.cc/2024/Conference — Submitted to ICLR 2024_

### Official Review · Reviewer_Z7u6 · 2023-10-31

**Soundness:** 3 good
**Presentation:** 3 good
**Contribution:** 3 good
**Rating:** 6
**Confidence:** 1

**Summary:**

This paper proposes a novel method named HYBRID for extracting maximally informative and minimally redundant high-order relationships from fMRI data. The authors argue that most current methods fail to accurately characterize interactions among brain regions because they only focus on pairwise connections and overlook high-order relationships. HYBRID addresses this limitation by constructing a hypergraph where hyperedges represent high-order relationships and their weights represent the strengths of those relationships. The authors demonstrate the effectiveness of HYBRID through comprehensive experiments, outperforming the state-of-the-art predictive model by an average of 12.1%. The contributions of this paper include a novel method for extracting high-order relationships from fMRI data and a comprehensive evaluation of the proposed method.

**Strengths:**

Originality: The paper proposes a novel method named HYBRID for extracting maximally informative and minimally redundant high-order relationships from fMRI data. This is a significant contribution as most current methods focus on pairwise connections and overlook high-order relationships. HYBRID addresses this limitation by constructing a hypergraph where hyperedges represent high-order relationships and their weights represent the strengths of those relationships. The proposed method is original and creative, and the authors provide a comprehensive evaluation of the proposed method, demonstrating its effectiveness through experiments.

Quality & clarity: The paper presents a clear problem formulation, a detailed description of the proposed method, and a comprehensive evaluation of the proposed method. The authors provide theoretical guarantees for the proposed method and demonstrate its effectiveness through experiments. The paper is well-written, with clear and concise language, making it easy to understand.

Significance: The paper addresses an important problem in neuroscience and machine learning. Discovering reliable and informative interactions among brain regions from fMRI signals is essential in neuroscientific predictions of cognition. Most of the current methods fail to accurately characterize those interactions because they only focus on pairwise connections and overlook the high-order relationships of brain regions. The proposed method addresses this limitation and provides a new approach for extracting high-order relationships from fMRI data.

**Weaknesses:**

The matrix notation in SEction 4 is confusing:
Section 4.1, first line: $X = [X_1, X_2, \dots, X_N]$ should be $X = [X_1, X_2, \dots, X_N]^T$ (i.e., a transpose operator should be inserted).
Eq. (4): a transpose operator should be inserted after the square brackets.

**Questions:**

Can you briefly explain how to choose the number of hyperedges?

---

> ### Author Response · Authors · 2023-11-15
> **Official Comments by Authors**
>
> We would like to address your concerns and questions below
> ## Weakness
>
> ### C1
>
> A transpose operator should be inserted.
>
> ### RC1
>
> Thanks for pointing this out and sorry for the confusion caused by the notation. We have inserted the transpose operator and gone through all the notations to make them more consistent and clear.
>
> ### Questions
> ### Q1
>
> Can you briefly explain how to choose the number of hyperedges?
>
> ### RQ1
>
> Sure! As stated in Section 4.1 (hyperedges construction), the number of hyperedges $K$ is a pre-defined hyperparameter. We choose $K=32$ based on the discussion and ablation studies that have been included in Appendix E in the submission. Basically, the performances improve as the $K$ goes larger, and becomes saturated after $K=32$.

---

### Official Review · Reviewer_yeBt · 2023-11-01

**Soundness:** 3 good
**Presentation:** 3 good
**Contribution:** 3 good
**Rating:** 6
**Confidence:** 3

**Summary:**

This work proposes a principle to learn high-order relationships of brain regions -- high-order relationships should be maximally informative and minimally redundant (MIMR), and a method called Hypergraph of Brain Regions via multi-head Drop- bottleneck (HyBRiD) to learn such relationships from fMRI data. HyBRiD includes a constructor to identify hyperedge structures, and a weighter to compute a weight for each hyperedge. The results show that HyBRiD outperformed 8 baseline methods in 7 out of 8 fMRI datasets.

**Strengths:**

1. The paper is well written with clear organization, detailed theoretical explanation, and comprehensive empirical evaluation.

2. The proposed method is intuitively simple yet effective, and could be potentially applied to learn high-order relationships of brain regions with respect to different prediction targets.

3. The neuroimaging experiments were comprehensive. A large sample size (8 datasets with 11875 subjects) was used to evaluate the models. The proposed method HyBRiD was compared to 3 types of baseline methods including 8 methods. HyBRiD outperformed 8 baseline methods in 7 out of 8 datasets.

4. The hyperedge profile analysis indicates interactions of multiple brain regions are more important in cognition tasks. The region importance reveals reasonable task-related brain regions under different conditions.

**Weaknesses:**

1. The authors mentioned that "Due to the data scarcity, training on individual datasets would result in serious overfitting." Each individual dataset includes at least 1000 subjects but the model performance is still not ideal. Can the model be applied to a dataset with less samples? Most clinical datasets are relatively small with < 1000 subjects. Is it applicable to apply the model on datasets with fewer samples?

2. It would be helpful if the authors could discuss potential reasons why HyBRiD failed at Rest 1 dataset.

3. Region importance. What about region importance for resting state data? If I understand it correctly, region importance is a metric for nodes. What about edges? Can you show how edges are connected under different conditions?

4. I appreciate that the authors include the code in supplemental material, but a README file should be also included to explain how to replicate the results.

5. The notations in Section 5.1 Metric are not very clear to me.

	i. The input of CPM should include the prediction target Y, right? Maybe use CPM(E, Y)?

	ii. What is the dimension of E? Is E equivalent to H+$\mathbf{w}$ in HyBRiD?

	iii. CPM is evaluated on each model separately, right? If so, I think Eq. 13 should be defined separately for HyBRiD.

6. Minor:

	i. Typos: Section 2.1 "Inupt" should be "Input"; Section 5.2 "conducte" should be "conduct"; Figure 7: "grpahical" should be "graphical".

	ii. CPM should be defined in the abstract and introduction upon its first occurrence.

	iii. In Section 5.1 Dataset, RS (resting state) should be defined.

	iv. In Table 5, $\beta = 0.2$ instead of $0.3$ to be consistent with Section E.3?

**Questions:**

1. Does the model generalize well across conditions or out-of-sample data? For example, if a model is trained on resting state data, can it be applied to predict task data?

---

> ### Author Response · Authors · 2023-11-15
> **Official Comments by Authors - Major**
>
> We would like to address your major concerns in this thread.
>
> ### C1
>
> The reviewer wondered if the model could be applied to a dataset with fewer samples.
>
> ### RC1
>
> While we do think training on a small dataset is an interesting question, we argue that both our model and the state-of-the-art one are unable to solve it, and finding out the solution to such a problem is out of the scope.
>
> To show it, we conducted experiments of training on individual datasets for both our model and BrainNetTF. The $r$ values:
>
> |            | SST 1  | EN-back 1 | MID 1 | Rest 1 |
> | ---------- | ------ | --------- | ----- | ------ |
> | BrainNetTF | -0.029 | 0.091     | 0.075 | 0.081  |
> | HyBRiD     | 0.103  | 0.085     | 0.002 | 0.060  |
>
> The $P$-values corresponds to these $r$ values:
>
> |            | SST 1 | EN-back 1 | MID 1 | Rest 1 |
> | ---------- | ----- | --------- | ----- | ------ |
> | BrainNetTF | 0.650 | 0.152     | 0.235 | 0.204  |
> | HyBRiD     | 0.105 | 0.178     | 0.975 | 0.341  |
>
> From this table, we can see all of the $P$-values> 0.05. Therefore, none of the existing models obtain **meaningful** results in this situation.
>
> However, our work implies that if you have small fMRI datasets, you can usually **combine** them for a better result. We hope this could be a good solution to the small datasets issue.
>
> ### C2
>
> Should discuss potential reasons why HyBRiD failed at Rest 1 dataset.
>
> ### RC2
>
> Thanks for your suggestion. We inspect the performance on both the training, validation and test datasets. We summarize the performances of our model and the SOTA one in the table below
>
> | model      | train | validation | test  |
> | ---------- | ----- | ---------- | ----- |
> | BrainNetTF | 0.967 | 0.484      | 0.334 |
> | HyBRiD     | 0.984 | 0.491      | 0.223 |
>
> We find our model outperforms BrainNetTF on both training and validation datasets but fails on test datasets. Therefore, given that Rest 1 is the **most noisy** dataset, we argue that the failure is because
>
> 1. Our model is more **expressive** than BrainNetTF, so it *overfits* the training dataset more than BrainNetTF.
> 2. The validation score is higher, indicating the failure reason is the **discrepancy** between the training dataset and the test dataset.
>
> Additionally, Note that although HyBRiD is not the best on Rest 1 dataset, it is still the **second-best** one.
>
> ### C3
>
> The reviewer would like to see region importance for resting state and individual hyperedges.
>
> ### RC3
>
> We did not visualize it in the original submission because different from task states, where specific brain regions are activated in response to particular tasks, brain activities during resting states, are not driven by external tasks, leading to more **diffuse and less predictable** patterns of activation. This makes it harder to interpret. However, we appreciate the reviewer’s interest and we have included the visualizations and interpretations of *resting-state region importance* and *individual hyperedges* in **Appendix G** in the updated version. We would like to do more analysis and interpretations in the future.
>
> ### Q1
>
> Does the model generalize well across conditions or out-of-sample data? For example, if a model is trained on resting state data, can it be applied to predict task data?
>
> ### RQ1
>
> Thanks for your interest in the model's generalizability. In our experience, training on resting-state and generalizing it on task-state is difficult because resting-state data is more **noisy**. However, we do think it is a good idea so instead, we train our model on task-state data and generalize it one resting-state data.
>
> |        | Training on all states, predicting on resting-state | Training on task-states, predicting on resting-state |
> | ------ | --------------------------------------------------- | ---------------------------------------------------- |
> | Rest 1 | 0.223                                               | 0.253                                                |
> | Rest 2 | 0.730                                               | 0.475                                                |
>
> The tables illustrate that our model demonstrates **a certain level of generalization**. For Rest 1, it is even better than training on Rest 1. We argue that it is because the model easily overfits on Rest 1 dataset, which is consistent with our response for **RC2.**

---

> ### Author Response · Authors · 2023-11-15
> **Official comments by Authors - Minor**
>
> We would like to address your minor concerns in this thread.
>
> ### C4
>
> Add a README file to explain how to replicate the results.
>
> ### RC4
>
> Thanks for the suggestion! We created an anonymous Github link https://anonymous.4open.science/r/HyBRiD-A7CC. Note that the code may be a little bit messy because we haven’t had time to organize and refactor it. We will do this as soon as possible in the future.
>
> ### C5
>
> Notations in Section 5.1 are not very clear.
>
> ### RC5
>
> i) I agree that CPM(E, Y) is more accurate and Eq. 13 has been updated.
>
> ii) $E$ denotes traditional pairwise edge weights, and it is a vector of length $K_p$, which is the total number of pairwise edges. We made it clearer by changing the notation $E$ to $\boldsymbol{W}_p$.
>
> iii) We made it clearer by changing $\boldsymbol{w}$ to  $\boldsymbol{w}_h$ for Eq.13. And made $\boldsymbol{w}_h = \boldsymbol{w}$ as a special case when evaluating on our model.
>
> You can check Eq.12 and Eq.13 in the updated version.
>
> ### C6
>
> i. Typos: Section 2.1 "Inupt" should be "Input"; Section 5.2 "conducte" should be "conduct"; Figure 7: "grpahical" should be "graphical".
>
> ii. CPM should be defined in the abstract and introduction upon its first occurrence.
>
> iii. In Section 5.1 Dataset, RS (resting state) should be defined.
>
> iv. In Table 5, $\beta=0.2$ instead of 0.3 to be consistent with Section E.3?
>
> ### RC6
>
> i) Thanks for pointing out. Typo fixed.
>
> ii) Thanks for the suggestion. We acknowledge the importance of defining 'CPM' upon its first appearance in the text. However, due to the constraints of the abstract and introduction sections, providing a full definition is challenging. We have, therefore, incorporated a concise explanation, stating that CPM is the evaluation protocol for our method, which gives readers an adequate initial background. In addition, we also refer to the appendix. We have updated:
>
> In abstract
>
> > Our model outperforms the state-of-the-art predictive model by an average of 12.1%, regarding the quality of hyperedges measured by CPM, a standard protocol for studying brain connections.
>
> In introduction
>
> > We quantitatively evaluate our approach by a commonly used protocol
> >
> >
> > for studying brain connections, CPM (Shen et al., 2017) (Appendix A), and show that…
>
> iii) Sorry for the confusion. We have modified it to
>
> > the resting state where the brain is not engaged in any activity (Rest)
>
> iv) Thanks for pointing out. Typo fixed.

---

### Official Review · Reviewer_eSJE · 2023-11-02

**Soundness:** 2 fair
**Presentation:** 2 fair
**Contribution:** 2 fair
**Rating:** 3
**Confidence:** 4

**Summary:**

The paper attempts at capturing multivariate relationships among a set or random variables as would be captured by edges in a hypergraph representation. For that, the paper constructs a differentiable regression model that first applies a set of learnable square (for the number of regions) linear projection with subsequent thresholding of the output - a mask, then applies an MLP to compute a scalar value (weight) for each of the masked subsets of the nodes. The weights are used to produce a scalar value (after an inner product with the output weight vector). This model is trained in a regularized regression manner and the produces features are evaluated in an acceptable feature selection evaluation pipeline using predictive strengths of the features as final evaluations. The approach is applied to a subset of the ABCD dataset.

**Strengths:**

The paper contains an interesting approach to model building, where an encoder builds clustering - a representation interpretable to human experts. A glass-layer with the partition clearly visible. Potentially, a rewrite of the paper could focus on this part, instead of the unsubstantiated claims about capturing high-order interations

**Weaknesses:**

1.  **The positioning of the paper is a problem.** The assertion that it captures high-order interaction is not substantiated, even though the feature selection model's entire motivation hinges on this claim. Certainly, the title emphasizes high-order interaction. However, the exact type of high-order interaction that the proposed model captures remains ambiguous. I would suggest considering the following papers, which were mistakenly overlooked. These papers seek to formally define what is being captured before attempting to estimate the interactions:
    -   Rosas FE, Mediano PA, Gastpar M, Jensen HJ. [Quantifying high-order interdependencies via multivariate extensions of the mutual information](https://journals.aps.org/pre/abstract/10.1103/PhysRevE.100.032305). Physical Review E. 2019 Sep 13;100(3):032305.
    -   Varley TF, Pope M, Faskowitz J, Sporns O. [Multivariate information theory uncovers synergistic subsystems of the human cerebral cortex](https://www.nature.com/articles/s42003-023-04843-w). Communications biology. 2023 Apr 24;6(1):451.
    -   Santoro A, Battiston F, Petri G, Amico E. [Higher-order organization of multivariate time series](https://www.nature.com/articles/s41567-022-01852-0). Nature Physics. 2023 Feb;19(2):221-9.
2.  **The clarity of the writing, with regard to the implementation**, is also inadequate in other sections. If we are estimating a hypergraph, then the edges, or node clusters, should form a cover rather than a partition. However, the regularization of the Mean Squared Error (MSE) used in Equation 11, as well as a preceding statement, both confirm the need for the edges to be disjoint, thereby suggesting a partition. This leads us back to the issue of positioning as it means that what is proposed is a clustering algorithm or the task of finding a partition. It's worth noting that node partitioning can still be conducted to recover high-order interactions, as has been exemplified in the neural imaging context, for instance, here:
    -   Plis SM, Sui J, Lane T, Roy S, Clark VP, Potluru VK, Huster RJ, Michael A, Sponheim SR, Weisend MP, Calhoun VD. [High-order interactions observed in multi-task intrinsic networks are dominant indicators of aberrant brain function in schizophrenia.](https://www.sciencedirect.com/science/article/pii/S1053811913007970) NeuroImage. 2014 Nov 15;102:35-48.
3.  Overall, **the approach feels like an ad hoc method** for grouping the feature vectors via their predictive potential for a dependent variable. Even the input features are correlation coefficients. That is, the initial input matrix for each subject is the correlation matrix of which the goal is to subselect rows (or columns, which is equivalent due to symmetry) into K different groups.
4.  **Comparisons in Table 1 are highly problematic** as well.
    1.  If the goal is to find high-order relations what does predictive quality of representations has to do with it? Table 1 in my opinion does not belong in a paper on high-order relations.
    2.  However, if the paper would be rewritten to focus on feature grouping and clustering, this approach may potentially work although not without changes. In this case, the proposed model needs to be compared with other approaches that do clustering or partition of random variables. For example, it seems appropriate to consider comparison with Deep Clustering.
5.  **Results are confusing.** They do not show individual "hyperedges" and analyze the ROIs that have grouped together and explain high-order interactions that grouped them. If the regularization enforces the partition, why are so many clusters overlap per my interpretation of Figures 5 and 4c?

**Questions:**

Note, I do not think answering my question below can change the problems with the way it is written which lead to experiments not supporting the claims. My comments below are to help future clarity of the work:

1.  The abstract states that CRM measures quality of hyperedges, however, CRM looks like a feature selection protocol that also has recommendations for assessing predictivity of the features. This is a disconnect between what is claimed and what is presented.
2.  The beginning of the second paragraph of the Introduction needs references to support the claim. The last phrase of that paragraph needs a rewrite "of the intricate behind brain regions"
3.  Section 2.1 "Inupt" - Do you mean Input?
4.  Section 2.2 describes feature selection - why is this tied to high-order interactions. Confusing.
5.  Section 4.1 what do you mean by "N-dimensional shallow embedding layer parameterized by". It would be best if all operations and parameters were clearly defined. I assume this is a linear transformation but that is a guess.
6.  Hyperedge weighting. MLP is not a sufficient description of the used model. Please also mention the activation function.
7.  Baselines: "standard" method mentioned there is unclear. What is standard? How is it defined?
8.  Why the model is compared with arbitrary models that solve problems different from the proposed method? How were models shown in the comparison selected?
9.  Ktena, Li, and Kan papers do not construct hypergraphs and yet used as such in the paper for comparisons. Confusing.
10. Consider fixing capitalization in your bib file. To do that, you can go over the cited papers in your .bib file and put all words you want to preserve capitalization of in additional curly braces. Like {fMRI}.

---

> ### Author Response · Authors · 2023-11-15
> **Official Comments by Authors - clarify misunderstandings**
>
> We believe there is a fundamental misunderstanding between the reviewer and us, which will be described below. In addition, we have also updated the paper to avoid other potential confusion. We would like to resolve the misunderstanding in this thread.
>
> ### C2
> The proposed method learns a partition, rather than a cover.
>
> ### RC2
> We would like to politely point out that the claim is incorrect. Our method indeed learns a **cover**, rather than a partition.
>
> We guess the reviewer obtained such a conclusion because the statement in the submission
>
> > The equality holds if and only if nodes are independent and hyperedges do not overlap
>
> However, this is a fundamental misunderstanding of our **optimization objective**. This sentence just stated the condition in which equality holds in this inequality.
>
> Specifically, Eq.10 shows
>
> $$
> I(H; X) \leq I_{\text{upper}}(H;X)
> $$
>
> where $I_\text{upper}(H;X)$ corresponds the right-hand-side of Eq.10. We optimize $I_\text{upper}(H;X)$ as a surrogate objective of  $I(H;X)$. In other words, we minimize $I(H;X)$ through minimizing $I_\text{upper}(H;X)$. This is a *common technique* in an abundance of works (e.g. in ELBO [1][2], in IB [3][4], and others [5][6]). Hence, the objective does not indicate partition.
>
> On the contrary, if one wants to force the partition, he/she should optimize the **tightness** instead
>
> $$
> T(H, X) = I_{\text{upper}}(H;X) - I(H; X)
> $$
>
> where $T(H,X)$ denotes the tightness and it is minimized when hyperedges do not overlap. And minimizing the tightness is not what we are doing in this work.
>
> Furthermore, the MSE term is **not for regularization**, it is the lower bound of informativeness in Eq.9, where $Y$ is the cognition score (defined in Section 2), which ensures the predictive performance of our high-order relations. The $I_{\text{upper}}$ actually corresponds to $\sum_{k=1}^K \sum_{i=1}^N \mathbb{H}[X_i](1-p_{\theta, i}^k)$ in Eq,10 & Eq.11, where one can find there is no force to partition at all.
>
> The property that a node can be connected by **any number** (0 to $K$) of hyperedges is attributed to our masking mechanism. Each hyperedge is generated by a learnable mask independently, which is free to connect to the same node.
>
> [1] Kingma, D. P., & Welling, M. (2013). Auto-encoding variational bayes. *arXiv preprint arXiv:1312.6114*.
>
> [2] Ho, J., Jain, A., & Abbeel, P. (2020). Denoising diffusion probabilistic models. *Advances in neural information processing systems*, *33*, 6840-6851.
>
> [3] Kim, J., Kim, M., Woo, D., & Kim, G. (2021). Drop-bottleneck: Learning discrete compressed representation for noise-robust exploration. *arXiv preprint arXiv:2103.12300*
>
> [4] Wu, T., Ren, H., Li, P., & Leskovec, J. (2020). Graph information bottleneck. *Advances in Neural Information Processing Systems*, *33*, 20437-20448.
>
> [5] Hjelm, R. D., Fedorov, A., Lavoie-Marchildon, S., Grewal, K., Bachman, P., Trischler, A., & Bengio, Y. (2018). Learning deep representations by mutual information estimation and maximization. *arXiv preprint arXiv:1808.06670*.
>
> [6] Veličković, P., Fedus, W., Hamilton, W. L., Liò, P., Bengio, Y., & Hjelm, R. D. (2018). Deep graph infomax. *arXiv preprint arXiv:1809.10341*.
>
> ### C4.2
>
> The proposed method should be compared with Deep Clustering if it will be rewritten to focus on feature grouping.
>
> ### RC4.2
>
> This concern is also a result of the misunderstanding of the model objective. Even if our task might be related to Deep Clustering, it differs a lot from the following aspects.
>
> 1. Our methods can learn a cover, which means a node can be connected to multiple hyperedges at the same time. However, most clustering methods usually assume each node only belongs to one cluster.
> 2. Most Deep Clustering methods usually need the supervision of cluster labels or semi-supervision signals of “must-links” and “cannot-links” [7].  However, we don’t have **such labels** besides the prediction labels.
> 3. Deep Clustering methods don’t characterize the clusters with a **succinct summary**. However, our methods compute a weight for each hyperedge, which is **informative** toward cognition.
> 4. In contrast to clustering methods, which usually assign each node to a specific cluster, our approach selectively **prunes** nodes that are not crucial for cognition. For example, for the EN-back condition, only 64 out of 164 regions are connected by at least one hyperedge.
>
> [7] Ren, Y., Pu, J., Yang, Z., Xu, J., Li, G., Pu, X., ... & He, L. (2022). Deep clustering: A comprehensive survey. *arXiv preprint arXiv:2210.04142*
>
> ### C5.2
>
> Clusters overlap with each other, which contradicts the regularization
>
> ### RC5.2
>
>  This is also a result of the misunderstanding in Q2. Our approach does not incorporate any regularization term to enforce the partition according to Eq.11 in the submission, so hyperedges are free to overlap with each other as long as the **overlap enhances the predictive performance**.

---

> ### Author Response · Authors · 2023-11-15
> **Official Comments by Authors - other concerns**
>
> We would like to address the reviewer's concerns in this thread.
>
> ## Weakness
> ### C1
>
> The author needs to clearly and formally define what type of high-order relations the proposed model captures.
>
> ### RC1
>
> We would like to politely point out that the high-order relations have been **intuitively** defined in the second paragraph of the Introduction (maximally informative and minimally redundant, MIMR) and **formally** defined in Equation 10 (IB principle).
>
> Specifically, our high-order relations are those that are most **predictive**, yet contain the **least** redundant information towards cognition. We have made the definition more clear and explicit in the updated version.
>
> The definition of ours is different from that in the three papers, where they consider nodes in a high-order relation should be collectively correlated in terms of co-fluctuations or some metrics like O-information.
>
> However, we appreciate the references you provided. We have included them and discussed the difference between their methods and ours in **Related Work (high-order relationships in fMRI)** in the updated version: Some of them are not scalable to high-order relations of a large degree, while some of them are inconsistent with our MIMR objective.
>
> ### C3
>
> The approach feels like an ad hoc method for grouping the feature vectors.
>
> ### RC3
>
> We would like to point out that our method is beyond feature grouping because
>
> 1. A node is free to be connected by different hyperedges at the same time, which makes it different from grouping or clustering.
> 2. Our method summarizes each hyperedge into a scalar, which is itself informative without the original features.
> 3. Our method also shows consistent predictive performance improvements other than selecting significant feature groups.
> 4. Our objective (MIMR) is based on a widely used principle and can be generalized on various tasks and various inputs.
>
> Although in practice we choose the Pearson correlation as our input features, our method can generalize on other features (e.g. time signals). Our choice of Pearson correlation coefficients as input features is intended to enable a **direct comparison** with the state-of-the-art method [9], and they use Pearson correlations as their node features.
>
> [9] Kan, Xuan, et al. "Brain network transformer." *Advances in Neural Information Processing Systems* 35 (2022): 25586-25599
>
> ### C4.1
>
> Predictive quality has nothing to do with discovering high-order relations.
>
> ### RC4.1
>
> This concern is a result of the misunderstanding of the model objective, as explained in our responses to Q1 and Q2, the high-order relations we want to discover in this work are the most **predictive** ones with minimal redundancy.
>
> ### C5.1
>
> The author does not show individual hyperedges.
>
> ### R5.1
>
> We focus on the region importance instead of individual hyperedges because we believe statistics make it clearer to present information. However, we appreciate your interest in individual hyperedges so we have included visualization and analysis of the most significant hyperedge under the EN-back condition in Appendix G.

---

> > ### Author Response · Authors · 2023-11-15
> > **Official Comments by Authors - questions**
> >
> > We would like to answer the reviewer's questions in this thread
> >
> > ### Questions
> >
> > 1. CPM looks like a feature selection protocol.
> >
> >    We would like to politely point out that CPM is not a feature selection protocol. Feature selection is only **one of the steps** in the model. Here’s the justification why we use it as an evaluation protocol for our method:
> >
> >    CPM was originally designed as a predictive model that predicts neurological outcomes based on edge weights. The evaluation of CPM could be measured by the correlation between predicted and ground-truth outcomes. The correlation is equal to the correlation between the sum of edges and the ground-truth outcome (The correlation is exactly the $r$ in Eq. 13) because CPM assumes **linear dependency** between the sum and the outcome. Hence, **one can expect a larger correlation if the edge weights are more correlated to the outcomes (and thus are more predictive)**.
> >
> > 2. (1) Add references at the beginning of the second paragraph and (2) rewrite "of the intricate behind brain regions”
> >
> >    (1) Thanks for the suggestion. References added.
> >
> >    (2) Rewrite the sentence to
> >
> >    > hinders the accurate understanding of brain region interactions (Cao et al., 2022; Richards et al., 2019)
> >
> > 3. Thanks for pointing out. Typo fixed
> >
> > 4. Section 2.2 describes feature selection, which is not tied to high-order relations.
> >
> >    Section 2.2 does not describe feature selection, it describes the background of the metric in Eq. 13, which uses CPM to evaluate the predictive performances of our learned hyperedges. For a more coherent presentation, we have moved all contents about CPM into Appendix B.
> >
> > 5. What does the author mean by "N-dimensional shallow embedding layer”. The reviewer assumes it is a linear transformation.
> >
> >    A shallow embedding layer, such as the token embedding layer commonly used in natural language processing (referred to in PyTorch as `torch.nn.Embedding`), is a simple lookup table that stores embeddings of a fixed dictionary and size. For better comprehension, we have omitted the term and instead provided a direct explanation:
> >
> >    > …,  where $p^k_{\theta, i}, i=1,2,\cdots, N$ are learnable parameters.
> >
> > 6. The author should mention the activation function in MLP
> >
> >    Thanks for the suggestion. The activation function is ReLU. We have updated it in the paper.
> >
> > 7. “standard” method mentioned is unclear.
> >
> >    “standard” is defined in the **Baselines** paragraph in Section 5.1 in the original submission
> >
> >    > standard method, which is exactly the classical pairwise-based one.
> >
> >    we have added references for such methods and made the definition clearer in the paper.
> >
> >    > standard method, which is exactly the classical standard method, which is exactly the classical method that predicts outcomes based on pairwise edges (Shen et al., 2017)
> >
> > 8&9. Improper baseline selection
> >
> > This question is also a result of the **misunderstanding of the model objective**.  Since we are aiming to find **predictive** high-order relations. It is critical to compare with state-of-the-art methods that predict neurological outcomes.
> >
> > 1. The author should fix the capitalization in the bib file.
> >
> >    Thanks for pointing out. Fixed.

---

### Official Review · Reviewer_JozW · 2023-11-10

**Soundness:** 2 fair
**Presentation:** 3 good
**Contribution:** 1 poor
**Rating:** 3
**Confidence:** 5

**Summary:**

This work proposed a hypergraph inference method based on optimizing the predictive power (in the form of mutual information) of the selected ("connected by hyperedge") graph node towards certain labels and the redundancy term. The effectiveness of the inference method was evaluated based on an fMRI condition classification task with comparison to pair-wise connectivity estimation and connectivity-based cognition prediction methods and achieved superior performance.

**Strengths:**

Eq. 9 and 10 provide a useful solution for the MRMR-like feature selection problem.

**Weaknesses:**

Formulating the hyperedge inference problem into a feature selection (MRMR-like) problem is interesting, yet not valid, at least in the context of functional connectivity analysis. Regions that are predictive together towards a certain cognitive condition do not imply that they are functionally connected. It's actually easy to construct a case where two regions have very similar fMRI signals, indicating potential strong functional connectivity, but will not be considered as "hyperedge connected" in the presented model as their information is redundant for the prediction.

**Questions:**

1) How is the p-value calculated for the hyperedge in Fig. 4?
2) It is recommended to discuss how the DimReduction MLP could be trained with a large number of nodes.
3) Are the linear-head Fl shared across hyperedges or trained separately for each hyperedge?

---

> ### Author Response · Authors · 2023-11-15
> **Offical comments by Authors**
>
> The reviewer stated
>
> > The effectiveness of the inference method was evaluated based on an fMRI condition classification task with comparison…
>
> We would like to politely point out that our method is not evaluated based on the fMRI condition classification task. Instead, as mentioned in Section 2 and Section 5.1 (Dataset) in the submission, our method is evaluated based on cognition score prediction, which is a regression task.
>
> ## Weaknesses
> ### C1
> It is not valid to infer functional connectivity according to the predictive performance.
>
> ### RC1
>
> We believe that there is a misunderstanding on the motivation. Traditional methods [1][2] typically define functional connectivity based on **similarity or correlation** (like Pearson correlation, and mutual information). However, this kind of connectivity is **not informative** toward a neurological outcome (see the performance of standard pairwise baseline in Table 1 as evidence) and might fall short in providing insights relevant to the outcome. As a result, it is crucial to learn these connections taking the outcome into account like what has been done in [3][4]. Therefore, our work does not seek to identify the traditional connections.
>
> Instead, as mentioned in 3rd paragraph of the submission, the objective of our method is to discover high-order connections that are **predictive** toward a cognition score (a part of our **MIMR objective**). And for regions that are connected by a predictive high-order relation, the relations between these regions can be highly non-linear and complex, and far beyond any existing manually-designed metrics (e.g. Pearson correlation, mutual information).
>
> [1] Shen, X., Finn, E. S., Scheinost, D., Rosenberg, M. D., Chun, M. M., Papademetris, X., & Constable, R. T. (2017). Using connectome-based predictive modeling to predict individual behavior from brain connectivity. *nature protocols*, *12*(3), 506-518.
>
> [2] Boyle, R., Connaughton, M., McGlinchey, E., Knight, S. P., De Looze, C., Carey, D., ... & Whelan, R. (2023). Connectome‐based predictive modeling of cognitive reserve using task‐based functional connectivity. *European Journal of Neuroscience*, *57*(3), 490-510.
>
> [3] Kawahara, J., Brown, C. J., Miller, S. P., Booth, B. G., Chau, V., Grunau, R. E., ... & Hamarneh, G. (2017). BrainNetCNN: Convolutional neural networks for brain networks; towards predicting neurodevelopment. *NeuroImage*, *146*, 1038-1049.
>
> [4] Mahmood, U., Fu, Z., Calhoun, V. D., & Plis, S. (2021). A deep learning model for data-driven discovery of functional connectivity. *Algorithms*, *14*(3), 75.
>
>
> ## Questions
> ### Q1
>
> How is the p-value calculated for the hyperedge in Fig. 4?
>
> ### RQ1
>
> As mentioned in Section 5.3
>
> > CPM conducts a significance test on pairwise edges and hyperedges internally based on a linear regression model, and thus we can obtain a P-value for each hyperedge from the significance test.
> >
>
> At a high level, CPM internally calculates a correlation coefficient for each hyperedge with the cognition score. Under a classical statistical hypothesis test framework, the coefficient follows a distribution. The p-value is obtained from the distribution. For details, we have added a paragraph about it in Appendix B.
>
> ### Q2
>
> How the DimReduction MLP could be trained with a large number of nodes.
>
> ### RQ2
>
> We apply DimReduction MLP independently on each hyperedge. As specified in Eq. 6 in the submission, for a hyperedge we first average features of nodes in that hyperedge by ${\boldsymbol{m}^k}^T \boldsymbol{h}^k$ and obtain the hyperedge features in $\mathbb{R}^{d}$. After that, the DimReduction MLP compresses the dimensionality from $d$ to $1$ as a scalar weight for the hyperedge. The parameters of the DimReduction MLP are shared for every hyperedge and every dataset.
>
> ### Q3
>
> Are the linear-head Fl shared across hyperedges or trained separately for each hyperedge?
>
> ### RQ3
>
> As specified in Eq.7, the inputs of the linear-head $\mathcal{F}_l$ are scalar weights of all hyperedges. Consequently, the notion of shared vs. separate linear-head functions in terms of hyperedges does not apply in this context regarding hyperedges. However, $\mathcal{F}_l$ is shared across all datasets.

---

### Author Response · Authors · 2023-11-15

We thank the reviewers for their recognition and suggestions. In conclusion,

### Strengths:

1. Reviewer yeBt and Z7u6 both think our method is **effective**, supported by both detailed **theoretical analysis** and **comprehensive empirical experiments**.
2. Reviewer yeBt and Z7u6 both think our paper is written with **clear organization** and easy to follow.
3. Reviewer yeBt and Z7u6 both think our model has shown that our high-order relations are more **effective than traditional** pairwise connections through the result analysis.
4. Reviewer Z7u6 thinks our method is **novel and creative**.

### Misunderstandings:

We would like to politely point out that there are fundamental misunderstandings between us and Reviewer JozW, eSJE.

1. We believe Reviewer JozW misunderstood the task we performed and questioned the validity of our high-order relationships.

    To resolve the misunderstanding, we have included **more context** of our motivation and added **more references** to prove the validity of the problem we are solving. In addition, We gave a **clearer and more explicit definition** of our high-order relationships and updated it in the paper.

2. We believe Reviewer JozW misunderstands our MIMR objective. He/she claims we learn a partition for nodes, where a node can only be connected by one hyperedge.

    To resolve the misunderstanding, we further explained each term in our **objective** in our submission and justified why there is no such partition regularization.


### Weaknesses & Solutions:

1. Reviewer eSJE thinks the definition of our high-order relationships is not clear and provided some references as examples.

    We have made the definition in the original submission **clearer and more explicit** in the updated version.

    For the references the reviewer provided, we added an extra paragraph in Related Work to demonstrate the difference between their goals from ours.

2. Reviewer eSJU questioned the evaluation process of our method.

    We believe we solved the concern by explaining the evaluation protocol we use in detail (also updated in the paper). And we explained why evaluating the predictive power of our high-order relationships makes sense under the MIMR objective.

3. Reviewer eSJU and yeBt asked us to show individual hyperedges.

    We added the visualization of the most significant hyperedge under EN-back condition as a case study in Appendix G.

4. Reviewer yeBt is interested in the generalizability of our method on out-of-sample data.

    We conducted more experiments and showed that our method possesses **some generalization capabilities**.

5. Reviewer yeBt requires us to explain the failure reason on Rest 1.

    We showed performances on training/validation/test datasets and demonstrated that the failure may be attributed to the expressiveness of our model, which leads to overfitting on the dataset.


We encourage reviewers to respond if they have any further questions. We kindly request the reviewers to re-evaluate the manuscript, considering these enhancements, and possibly reassess their scores if they feel it is warranted.

---

### Meta-Review · Area_Chair_LicL · 2023-12-19

**Metareview:**

Discover higher-order relationships between brain regions that best predict fluid intelligence. This is an interesting idea that goes beyond functional connectivity and pairwise influence between brain regions.

Reviewers had a number of concerns:
- The authors rely only on results on the ABCD dataset (which they subset in various ways) to show the validity of their hyperedges. But, immediately, one will wonder if these hyperedges make any sense? How should one interpret a 20-degree hyperedge between brain regions? What other evidence is there for this higher order connection? The authors assert that all that matters is the statistics of the results, but this just isn’t the case, neuroscience seeks a level of understanding and validation that the manuscript doesn’t engage with. Note that even the authors say at times “Deep Clustering methods don’t characterize the clusters with a succinct summary. However, our methods compute a weight for each hyperedge, which is informative toward cognition.” This is really a necessity for publication, otherwise this is a set of potentially-interesting results without any context to understand them.
- The reviewers were skeptical of the idea that merely producing better predictions can be a vehicle for finding neurologically valid hyperedges, rather than just an artifact that helps the prediction task. Authors argue that their MIMR objective function is such that they will find the minimum-sized hyperedge; but there is no evidence for this. Two critical experiments are missing. One would be a simulated experiment with artificial data to show that the method does as the authors claim “our high-order relations are those that are most predictive, yet contain the least redundant information towards cognition”. Another critical step of establishing that these hyperedges are real, would be to run on multiple datasets, rather than just ABCD, and compare the stability of the hyperedges that are found.

The authors in a sense assume that readers will accept the idea that their MIMR objective works the way they claim it does. But this must be established rather than assumed.

Reviewers mentioned several other important points in addition to the above that I encourage the authors to fully engage with and find experiments and analyses to better demonstrate their approach.

This could be a good submission, but these critical issues of validity and convincing demonstrations of the method must be addressed. At present, most would have the same reaction as the reviewers of being skeptical of the resulting hyperedges.

**Justification For Why Not Higher Score:**

The basic validity of the method and approach must be established first.

**Justification For Why Not Lower Score:**

N/A

---

### Decision · Program_Chairs · 2024-01-16

Reject